# Why Deep Surgical Models Fail?: Revisiting Surgical Action Triplet Recognition through the Lens of Robustness

**Yanqi Cheng**[1]    **Lihao Liu**[1]    **Shujun Wang**[1]    **Yueming Jin**[2]    **Carola-Bibiane Schönlieb**[1]
**Angelica I. Aviles-Rivero**[1]

[1] Department of Applied Mathematics and Theoretical Physics, University of Cambridge;
  {`yc443, ll610, sw991, cbs31, ai323`}`@cam.ac.uk`
[2] Wellcome/EPSRC Centre for Interventional and Surgical Sciences and Department of
  Computer Science, UCL; `yueming.jin@ucl.ac.uk`

## Abstract

Surgical action triplet recognition provides a better understanding of the surgical scene. This task is of high relevance as it provides the surgeon with context-aware support and safety. The current go-to strategy for improving performance is the development of new network mechanisms. However, the performance of current state-of-the-art techniques is substantially lower than other surgical tasks. Why is this happening? This is the question that we address in this work. We present the first study to understand the failure of existing deep learning models through the lens of robustness and explainability. Firstly, we study current existing models under weak and strong $\delta-$perturbations via an adversarial optimisation scheme. We then analyse the failure modes via feature based explanations. Our study reveals that the key to improving performance and increasing reliability is in the core and spurious attributes. Our work opens the door to more trustworthy and reliable deep learning models in surgical data science.
https://yc443.github.io/robustIVT/

## 1 Introduction

Minimally Invasive Surgery (MIS) has become the gold standard for several procedures (i.e., cholecystectomy & appendectomy), as it provides better clinical outcomes including reducing blood loss, minimising trauma to the body, causing less post-operative pain and faster recovery (Velanovich, 2000; Wilson et al., 2014). Despite the benefits of MIS, surgeons lose direct vision and touch on the target, which decreases surgeon-patient transparency imposing technical challenges to the surgeon. These challenges have motivated the development of automatic techniques for the analysis of the surgical workflow (Aviles et al., 2016; Maier-Hein et al., 2017; Vercauteren et al., 2019; Nwoye et al., 2022). In particular, this work aims to address a key research problem in surgical data science—surgical recognition, which provides to the surgeon context-aware support and safety.

The majority of existing surgical recognition techniques focus on phase recognition (Blum et al., 2010; Dergachyova et al., 2016; Lo et al., 2003; Twinanda et al., 2016; Zisimopoulos et al., 2018). However, phase recognition is limited by its own definition; as it does not provide complete information on the surgical scene. We therefore consider the setting of *surgical action triplet recognition*, which offers a better understanding of the surgical scene. The goal of triplet recognition is to recognise the ⟨instrument, verb, target⟩ and their inherent relations. A visualisation of this task is displayed in Figure 1.

The concept behind triplet recognition has been recognised in the early works of that (Neumuth et al., 2006; Katić et al., 2014). However, it has not been until the recent introduction of richer datasets, such as CholecT40 (Nwoye et al., 2020), that the community started developing new techniques under more realistic conditions. The work of that Nwoye et al (Nwoye et al., 2020) proposed a framework called Tripnet, which was the first work to formally address surgical actions as triplets.

In that work, authors proposed a 3D interaction space for learning the triplets. In more recent work, the authors of Nwoye et al. (2022) introduced two new models. The first one is a direct extension of Tripnet called Attention Tripnet, where the novelty relies on a spatial attention mechanism. In the same work, the authors introduced another model called Rendezvous (RDV) that highlights a transformer-inspired neural network.

A commonality of existing surgical action triplet recognition techniques is the development of new mechanisms for improving the network architecture. However and despite the potential improvements, the performance of existing techniques is substantially lower than other tasks in surgical sciences—for example, force estimation and navigation assisted surgery. In this work, we go contrariwise existing techniques, and tackle the surgical action triplet recognition problem from the lens of robustness and explainability.

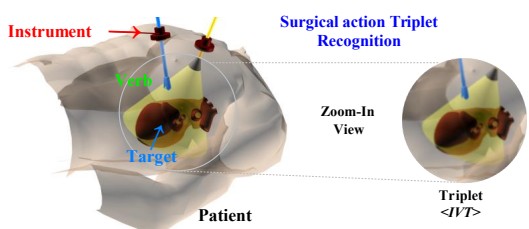

Figure 1: Visualisation of the surgical action triplet recognition task. We consider the tasks where the instrument ($I$), verb ($V$, action), and target ($T$, anatomical part) seek to be predicted.

In the machine learning community there is a substantial increase of interest in understanding the lack of reliability of deep learning models (e.g., Ribeiro et al. (2016); Koh & Liang (2017); Sundararajan et al. (2017); Liu et al. (2019); Yeh et al. (2019); Hsieh et al. (2020)). To understand the lack of reliability of existing deep networks, a popular family of techniques is the so-called *feature based explanations* via robustness analysis (Simonyan et al., 2013; Zeiler & Fergus, 2014; Plumb et al., 2018; Wong et al., 2021; Singla & Feizi, 2021). Whilst existing techniques have extensively been evaluated for natural images tasks, there are no existing works addressing the complex problems as in action triplet recognition.

☞ **Contributions.** In this work, we introduce, to the best of our knowledge, *the first study to understand the failure of existing deep learning models for surgical action triplet recognition.* To do this, we analyse the failures of existing state-of-the-art solutions through the lens of robustness. Specifically, we push to the limit the existing SOTA techniques for surgical action triplet recognition under weak and strong $\delta-$perturbations. We then extensively analyse the failure modes via the evaluation criteria Robustness-$S$, which analyses the behaviour of the models through feature based explanations. Our study reveals the impact of core and spurious features for more robust models. Our study opens the door to more trustworthy and reliable deep learning models in surgical data science, which is imperative for MIS.

## 2 METHODOLOGY

We describe two key parts for Surgical action triplet recognition task: i) our experimental settings along with assumptions and ii) how we evaluate robustness via adversarial optimisation. The workflow of our work is displayed in Figure 2.

### 2.1 SURGICAL ACTION TRIPLET RECOGNITION

In the surgical action triplet recognition problem, the main task is to recognise the triplet $IVT$, which is the composition of three components during surgery: instrument ($I$), verb ($V$), and target ($T$) in a given RGB image $x \in \mathbb{R}^{H \times W \times 3}$.

Formally, we consider a given set of samples $\{(x_n, y_n)\}_{n=1}^{N}$ with provided labels $\mathscr{Y} = \{0, 1, .., C_{IVT} - 1\}$ for $C_{IVT} = 100$ classes. We seek then to predict a function $f : \mathscr{X} \mapsto \mathscr{Y}$ such that $f$ gets a good estimate for the unseen data. That is, a given parameterised deep learning model takes the image $x$ as input, and outputs a set of class-wise presence probabilities, in our case 100 classes, under the $IVT$ composition, $Y_{IVT} \in \mathbb{R}^{100}$, which we call it the logits of $IVT$. Since there are three individual components under the triplet composition, within the training network, we also considered the individual component $d^* \in \{I, V, T\}$, each with class number $C_{d^*}$ (i.e. $C_I = 6$, $C_V = 10$, $C_T = 15$). The logits of each component, $Y_{d^*} \in \mathbb{R}^{C_{d^*}}$, are computed and used within the network.

In current state-of-the-art (SOTA) deep models (Nwoye et al., 2020; 2022), there is a communal structure divided into three parts: i) the feature extraction backbone; ii) the individual component encoder; and iii) the triplet aggregation decoder that associate the components and output the logits of the *IVT* triplet. More precisely, the individual component encoder firstly concentrates on the instrument component to output Class Activation Maps (CAMs $\in \mathbb{R}^{H \times W \times C_d}$) and the logits $Y_I$ of the instrument classes; the CAMs are then associated with the verb and target components separately for their logits ($Y_V$ and $Y_T$) to address the instrument-centric nature of the triplet.

The current SOTA techniques for surgical action triplet recognition focus on improving the components ii) & iii). However, the performance is still substantially lower than other surgical tasks. Our intuition behind such behaviour is due to the inherently complex and ambiguous conditions in MIS, which reflects the inability of the models to learn meaningful features. Our work is then based on the following modelling hypothesis.

> **Hypothesis 2.1: Deep Features are key for Robustness**
>
> *Deep surgical techniques for triplet recognition lacks reliability due to the ineffective features. Therefore, the key to boosting performance, improving trustworthiness and reliability, and understanding failure of deep models is in the deep features.*

Following previous hypothesis, we address the questions of—why deep triplet recognition models fail? We do that by analysing the feature based explanations via robustness. To do this, we consider the current three SOTA techniques for our study: Tripnet (Nwoye et al., 2020), Attention Tripnet, and Rendezvous (Nwoye et al., 2022). Moreover, we extensively investigate the repercussion of deep features using four widely used backbones ResNet-18, ResNet-50 (He et al., 2015), DenseNet-121 (Huang et al., 2016), and Swin Transformer (Liu et al., 2021). In the next section, we detail our strategy for analysing robustness.

Figure 2: Illustration of the main network structure, and how the adversarial perturbation is added to measure robustness.

## 2.2 FEATURE BASED EXPLANATIONS VIA ROBUSTNESS

Our models of the triplet recognition output the logits of triplets composition, we then use it to select our predicted label for the classification result. We define the model from image $x$ to the predicted label $\hat{y}$ as $f : \mathscr{X} \to \mathscr{Y}$, where $\mathscr{X} \subset \mathbb{R}^{H \times W \times 3}, \mathscr{Y} = \{0, 1, 2, ..., C_{IVT} - 1\}$.

For each class $m \in \mathscr{Y}$ and within each given sample, we seek to recognise core and spurious attributions (Singla & Feizi, 2021; Singla et al., 2021), which definition is as follows.

- **Core Attributes:** they refer to the features that form a part in the object we are detecting.
- **Spurious Attributes:** these are the ones that not a part of the object but co-occurs with it.

**How We Evaluate Robustness?** The body of literature has reported several alternatives for addressing the robustness of deep networks. Our work is motivated by recent findings on perturbation based methods, where even a small perturbation can significantly affect the performance of neural nets. In particular, we consider the setting of adversarial training (Allen-Zhu & Li, 2022; Olah et al., 2018; Engstrom et al., 2019) for *robustify* a given deep model.

The idea behind adversarial training for robustness is to enforce a given model to maintain its performance under a given perturbation $\delta$. This problem can be seen cast as an optimisation problem over the network parameters $\theta$ as:

$$\theta^* = \arg\min_{\theta} \mathbb{E}_{(\boldsymbol{x},y) \sim \mathscr{D}}[\mathscr{L}_{\theta}(\boldsymbol{x}, y)]. \tag{1}$$

where $\mathbb{E}[\mathscr{L}_\theta(\cdot)]$ denotes the expected loss to the parameter $\theta$.

One seeks to the model be resistant to any $\delta-$perturbation. In this work, we follow a generalised adversarial training model, which reads:

---

**Definition 2.1: Adversarial training under $\delta$**

$$\theta^* = \arg\min_\theta \mathbb{E}_{(\boldsymbol{x},y)\sim\mathscr{D}}[\max_{\boldsymbol{\delta}\in\boldsymbol{\Delta}} \mathscr{L}_\theta(\boldsymbol{x}+\boldsymbol{\delta},y)].$$

---

The goal is to the models do not change their performance even under the worse (strong) $\delta$.

The machine learning literature has explored different forms of the generalised model in definition equation 2.1. For example, a better sparsity regulariser for the adversarial training as in (Xu et al., 2018). In this work, we adopt the evaluation criteria of that (Hsieh et al., 2020), where one seeks to measure the susceptibility of features to adversarial perturbations. More precisely, we can have an insight of the deep features extracted by our prediction through visualising compact set of relevant features selected by some defined explanation methods on trained models, and measuring the robustness of the models by performing adversarial attacks on the relevant or the irrelevant features.

We denote the set of all features as $U$, and consider a general set of feature $S \subseteq U$. Since the feature we are interested are those in the image $\boldsymbol{x}$, we further denote the subset of $S$ that related to the image as $\boldsymbol{x}_S$. To measure the robustness of the model, we rewrote the generalised model equation 2.1 following the evaluation criteria of that (Hsieh et al., 2020). A model on input $\boldsymbol{x}$ with adversarial perturbation on feature set $S$ then reads:

---

**Definition 2.2: Adversarial $\delta$ & Robustness-$S$**

$$\varepsilon^*_{\boldsymbol{x}_S} := \{\min_{\boldsymbol{\delta}} \|\boldsymbol{\delta}\|_p \quad s.t. f(\boldsymbol{x}+\boldsymbol{\delta}) \neq y, \quad \boldsymbol{\delta}_{\overline{S}} = 0\},$$

---

where $y$ is the ground truth label of image $\boldsymbol{x}$; $\|\cdot\|_p$ denotes the adversarial perturbation norm; $\overline{S} = U \setminus S$ denotes the complementary set of feature $S$ with $\boldsymbol{\delta}_{\overline{S}} = 0$ constraining the perturbation only happens on $\boldsymbol{x}_S$. We refer to $\varepsilon^*_{\boldsymbol{x}_S}$ as **Robustness-$S$** (Hsieh et al., 2020), or the minimum adversarial perturbation norm on $\boldsymbol{x}_S$.

We then denote the relevant features selected by the explanation methods as $S_r \subseteq U$, with the irrelevant features as its complementary set $\overline{S_r} = U \setminus S_r$. Thus, the robustness on chosen feature sets—$S_r$ and $\overline{S_r}$ tested on image $\boldsymbol{x}$ are:

$$\text{Robustness-}S_r = \varepsilon^*_{\boldsymbol{x}_{S_r}}; \quad \text{Robustness-}\overline{S_r} = \varepsilon^*_{\boldsymbol{x}_{\overline{S_r}}}.$$

Table 1: Performance comparison for the task of Triplet recognition. The results are reported in terms of Average Precision (*AP%*) on the CholecT45 dataset using the official cross-validation split.

| METHOD | | COMPONENT DETECTION | | | TRIPLET ASSOCIATION | | |
|---|---|---|---|---|---|---|---|
| BASELINE | BACKBONE | $AP_I$ | $AP_V$ | $AP_T$ | $AP_{IV}$ | $AP_{IT}$ | $AP_{IVT}$ |
| Tripnet | ResNet-18 | $82.4\pm2.5$ | $54.1\pm2.0$ | $33.0\pm2.3$ | $30.6\pm2.6$ | $25.9\pm1.5$ | $21.2\pm1.2$ |
| | ResNet-50 | $85.3\pm1.3$ | $57.8\pm1.6$ | $34.7\pm1.9$ | $31.3\pm2.3$ | $27.1\pm2.4$ | $21.9\pm1.5$ |
| | DenseNet-121 | $86.9\pm1.4$ | $58.7\pm1.5$ | $35.6\pm2.8$ | $33.4\pm3.4$ | $27.8\pm1.8$ | $22.5\pm2.3$ |
| Attention Tripnet | ResNet-18 | $82.2\pm2.6$ | $56.7\pm3.8$ | $34.6\pm2.2$ | $30.8\pm1.8$ | $27.4\pm1.3$ | $21.7\pm1.3$ |
| | ResNet-50 | $81.9\pm3.0$ | $56.8\pm1.1$ | $34.1\pm1.4$ | $31.5\pm2.2$ | $27.5\pm1.0$ | $21.9\pm1.2$ |
| | DenseNet-121 | $83.7\pm3.5$ | $57.5\pm3.2$ | $34.3\pm1.3$ | $33.1\pm2.4$ | $28.5\pm1.6$ | $22.8\pm1.3$ |
| Rendezvous | ResNet-18 | $85.3\pm1.4$ | $58.9\pm2.6$ | $35.2\pm3.4$ | $33.6\pm2.6$ | $30.1\pm2.8$ | $24.3\pm2.3$ |
| | ResNet-50 | $85.4\pm1.6$ | $58.4\pm1.4$ | $34.7\pm2.4$ | $35.3\pm3.5$ | $30.8\pm2.6$ | $25.3\pm2.7$ |
| | DenseNet-121 | $88.5\pm2.7$ | $61.7\pm1.7$ | $36.7\pm2.1$ | $36.5\pm4.7$ | $32.1\pm2.7$ | $26.3\pm2.9$ |
| | Swin-T | $73.6\pm1.9$ | $48.3\pm2.6$ | $29.2\pm1.4$ | $28.1\pm3.1$ | $24.7\pm2.0$ | $20.4\pm2.1$ |

## 3 EXPERIMENTAL RESULTS

In this section, we describe in detail the range of experiments that we conducted to validate our methodology.

Table 2: Heatmaps Comparison under different feature extraction backbones. We displayed four randomly selected images in fold 3 when using the best performed weights trained and validated on folds 1,2,4 and 5.

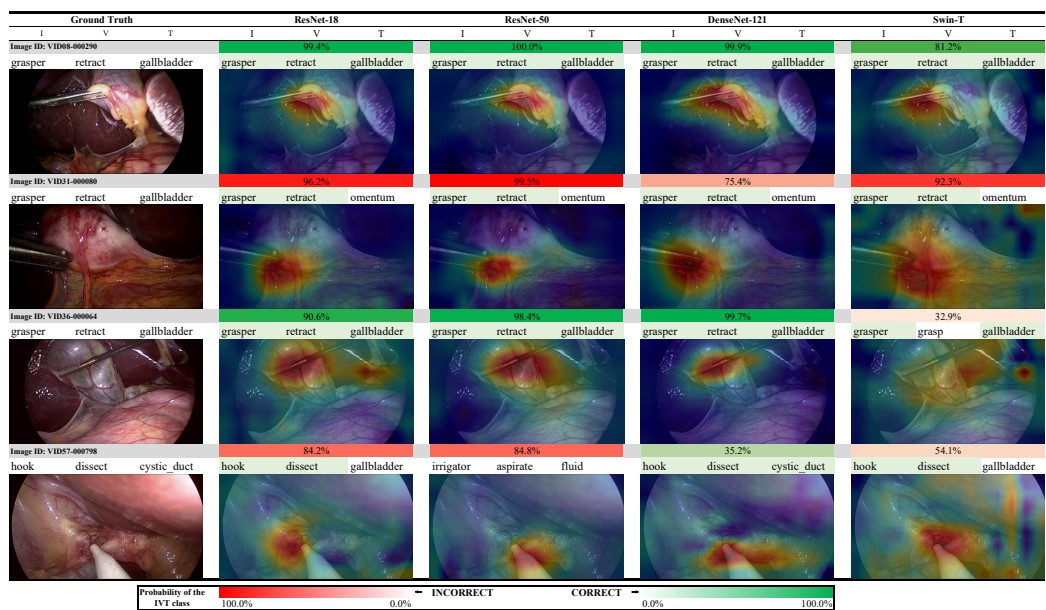

Table 3: Top 5 predicted Triplet classes in each of the 10 models. The top 5 is assessed by the $AP_{IVT}$ score.

| | ResNet-18 | | | | ResNet-50 | | | | DenseNet-121 | | | | Swin-T | | | |
|---|---|---|---|---|---|---|---|---|---|---|---|---|---|---|---|---|
| | Triplet | | | AP | Triplet | | | AP | Triplet | | | AP | | | | |
| Tripnet | 12:grasper | grasp | specimen_bag | 82.60% | 17:grasper | retract | gallbladder | 86.95% | 17:grasper | retract | gallbladder | 86.93% | | | | |
| | 17:grasper | retract | gallbladder | 81.04% | 12:grasper | grasp | specimen_bag | 80.50% | 12:grasper | grasp | specimen_bag | 81.45% | | | | |
| | 29:bipolar | coagulate | liver | 77.11% | 60:hook | dissect | gallbladder | 77.15% | 29:bipolar | coagulate | liver | 80.19% | | | | |
| | 60:hook | dissect | gallbladder | 74.13% | 29:bipolar | coagulate | liver | 75.69% | 60:hook | dissect | gallbladder | 76.35% | | | | |
| | 79:clipper | clip | cystic_duct | 61.28% | 6:grasper | grasp | cystic_plate | 69.24% | 79:clipper | clip | cystic_duct | 67.75% | | | | |
| | Triplet | | | AP | Triplet | | | AP | Triplet | | | AP | | | | |
| Attention Tripnet | 12:grasper | grasp | specimen_bag | 81.38% | 17:grasper | retract | gallbladder | 82.75% | 17:grasper | retract | gallbladder | 83.63% | | | | |
| | 17:grasper | retract | gallbladder | 78.70% | 12:grasper | grasp | specimen_bag | 78.53% | 12:grasper | grasp | specimen_bag | 80.01% | | | | |
| | 29:bipolar | coagulate | liver | 78.52% | 29:bipolar | coagulate | liver | 76.44% | 29:bipolar | coagulate | liver | 75.68% | | | | |
| | 28:bipolar | coagulate | gallbladder | 77.44% | 60:hook | dissect | gallbladder | 71.79% | 60:hook | dissect | gallbladder | 75.36% | | | | |
| | 30:bipolar | coagulate | omentum | 77.39% | 28:bipolar | coagulate | gallbladder | 70.68% | 30:bipolar | coagulate | omentum | 69.49% | | | | |
| | Triplet | | | AP | Triplet | | | AP | Triplet | | | AP | Triplet | | | AP |
| Rendezvous | 17:grasper | retract | gallbladder | 85.57% | 30:bipolar | coagulate | omentum | 91.36% | 84:irrigator | dissect | cystic_pedicle | 96.84% | 17:grasper | retract | gallbladder | 78.36% |
| | 29:bipolar | coagulate | liver | 83.90% | 17:grasper | retract | gallbladder | 86.11% | 30:bipolar | coagulate | omentum | 89.60% | 60:hook | dissect | gallbladder | 72.57% |
| | 12:grasper | grasp | specimen_bag | 82.77% | 29:bipolar | coagulate | liver | 84.94% | 17:grasper | retract | gallbladder | 89.46% | 12:grasper | grasp | specimen_bag | 69.96% |
| | 30:bipolar | coagulate | omentum | 76.88% | 12:grasper | grasp | specimen_bag | 81.50% | 12:grasper | grasp | specimen_bag | 85.88% | 30:bipolar | coagulate | omentum | 67.03% |
| | 60:hook | dissect | gallbladder | 76.49% | 28:bipolar | coagulate | gallbladder | 79.60% | 29:bipolar | coagulate | liver | 84.43% | 29:bipolar | coagulate | liver | 66.08% |

## 3.1 DATASET DESCRIPTION AND EVALUATION PROTOCOL

**Dataset Description.** We use CholecT45 dataset (Nwoye & Padoy, 2022) to evaluate the robustness of the three SOTA models for the Surgical Action Triplet Recognition task. Specifically, CholecT45 dataset contains 45 videos with annotations including 6 classes of instrument, 10 classes of verb, and 15 classes of target (i.e. $C_I = 6, C_V = 10, C_T = 15$) generating 900 ($6 \times 10 \times 25$) potential combinations for triplet labels. To maximise the clinical utility, we utilise the top-100 combinations of relevant labels, which are selected by removing a large portion of spurious combinations according to class grouping and surgical relevance rating (Nwoye et al., 2022). Each video contains around $2,000$ annotated frames extracted at 1 fps in RGB channels, leading to a total of $90,489$ recorded frames. To remove the redundant information, the frames captured after the laparoscope been taken out of the body are blacked out with value $[0,0,0]$.

**Evaluation Protocol.** The triplet action recognition is evaluated by the average precision ($AP$) metric. Our models can directly output the predictions of triplet class $AP_{IVT}$. Instead, $AP_d$ where $d \in \{I, V, T, IV, IT\}$ cannot be predicted explicitly. Then we obtain the final predictions of $d \in \{I, V, T, IV, IT\}$ components according to (Nwoye & Padoy, 2022; Nwoye et al., 2022):

$$Y_d^k = \max_m \{Y_{IVT}^m\}, \quad \forall m \in \{0, 1.., C_{IVT} - 1\} \, s.t. \, h_d(m) = k,$$

where we calculate the probability of class $k \in \{0, 1, .., C_d - 1\}$ under component $d$; and $h_d(\cdot)$ maps the class $m$ from $IVT$ triplet compositions to the class under component $d$.

In our robustness analysis, the main evaluation criteria is the robustness subject to the selected feature set ($S_r$ and $\overline{S_r}$) on each backbone using the formula in equation 2.2.

## 3.2 IMPLEMENTATION DETAILS

We evaluate the model performance based on five-fold cross-validation, where we split 45 full videos into 5 equal folds. The testing set is selected from these 5 folds, and we treat the remaining 4 folds as the training set. Moreover, 5 videos from the 36 training set videos are selected as validation set during training.

The models are trained using the Stochastic Gradient Descent (SGD) optimiser. The feature extraction backbones are initialised with ImageNet pre-trained weights. Both linear and exponential decay of learning rate are used during training, with initial learning rates as $\{1e^{-2}; 1e^{-2}, 1e^{-2}\}$ for backbone, encoder and decoder parts respectively. We set the batch size as 32, and epoch which performs the best among all recorded epochs up to $AP$ score saturation on validation set in the specified k-fold. To reduce computational load, the input images and corresponding segmentation masks are resized from $256 \times 448$ to $8 \times 14$. For fair comparison, we ran all SOTA models (following all suggested protocols from the official repository) under the same conditions and using the official cross-validation split of the CholecT45 dataset (Nwoye & Padoy, 2022).

## 3.3 EVALUATION ON DOWNSTREAM TASKS

In this section, we carefully analyse the current SOTA techniques for triplet recognition from the feature based explainability lens.

↻ **Results on Triplet Recognition with Cross-Validation.** As first part of our analysis, we investigate the performance limitation on current SOTA techniques, and emphasise how such limitation is linked to the lack of reliable features. The results are reported in Table 1. In a closer look at the results, we observe that ResNet-18, in general, performs the worst among the compared backbones. However, we can observe that for one case, component analysis, it performs better than ResNet-50 under Tripnet Attention baseline. The intuition being such behaviour is that the MIS setting relies on ambiguous condition and, in some cases, some frames might contain higher spurious features that are better captured by it. We remark that the mean and standard-deviation in Table 1 are calculated from the 5 folds in each combination of backbone and baseline.

We also observe that ResNet-50 performs better than ResNet-18 due to the deeper feature extraction. The best performance, for both the tasks—component detection and triplet association, is reported by DenseNet-121. The intuition behind the performance gain is that DenseNet-121 somehow mitigates the issue of the limitation of the capability representation. This is because ResNet type networks are limited by the identity shortcut that stabilises training. These results support our modelling hypothesis that the key of performance is the robustness of the deep features.

A key finding in our results is that whilst existing SOTA techniques (Nwoye & Padoy, 2022; Nwoye et al., 2022) are devoted to developing new network mechanisms, one can observe that a substantial performance improvement when improving the feature extraction. Moreover and unlike other surgical tasks, current techniques for triplet recognition are limited in performance. Why is this happening? Our results showed that the key is in the *reliable features* (linked to robustness); as enforcing more meaningful features, through several backbones, a significant performance improvement over all SOTA techniques is observed.

To further support our previous findings, we also ran a set of experiments using the trending principle of Transformers. More precisely, an non CNN backbone—the tiny Swin Transformer (Swin-T) (Liu et al., 2021) has also been tested on the Rendezvous, which has rather low $AP$ scores on all of the 6 components in oppose to the 3 CNN backbones. This could be led by the shifted windows in the Swin-T, it is true that the shifted windows largely reduced the computational cost, but this could lead to bias feature attribute within bounding boxes, the incoherent spreading can be seen clearly in the visualisation of detected relevant features in Swin-T in Figure 3 (a).

In Table 1 we displayed the average results over all classes but—what behaviour can be observed from the per-class performance? It can be seen from Table 3 that though the best 5 predicted classes are different in each model, the predicted compositions seem clinically sensible supporting our previous discussion. In addition, the top 1 per-class *AP* score is significantly higher in DenseNet-121 with Rendezvous.

↻ **Visualisation Results.** To interpret features is far from being trivial. To address this issue, we provide a human-like comparison via heatmaps in Table 2. The implementation of the heatmaps is adapted from (Zhou et al., 2016). The displayed outputs reflect what the model is focusing based on the extracted features. These results support our hypothesis that deep features are the key in making correct predictions over any new network mechanism.

We observed that in the worst performed backbone—Swin-T, the feature been extracted are mostly spread across the images, however, the ones that concentrate on core attributes are not though performed the best. In the best performed DenseNet-121, a reasonable amount of attention are also been paid to spurious attributes; this can be seen more directly in our later discussion on robustness visualisation Figure 3.

The reported probability on the predicted label emphasises again the outstanding performance of DenseNet-121 backbone; in the sense that, the higher the probability for the correct label the better, the lower it is for incorrect prediction the better.

↻ **Why Surgical Triplet Recognition Models Fail? Robustness and Interpretability.** We further support our findings through the lens of robustness. We use as evaluation criteria Robustness-$S_r$ and Robustness-$\overline{S_r}$ with different explanation methods: vanilla gradient (Grad) (Shrikumar et al., 2017) and integrated gradient (IG) (Sundararajan et al., 2017). The results are in Table 4 & Figure 3.

Table 4: Robustness measured on 400 examples (i.e. images) randomly selected from the images in the fold 3 videos with exactly 1 labeled triplet. Top 25 percent of relevant $S_r$ or irrelevant $\overline{S_r}$ features are selected from 2 explanation methods Grad and IG. We perform attacks on the selected 25 percent.

| ATTACKED FEATURES | EXPLANATION METHODS | BACKBONES (ON RENDEZVOUS) | | | |
| --- | --- | --- | --- | --- | --- |
| | | ResNet-18 | ResNet-50 | DenseNet-121 | Swin-T |
| Robustness-$\overline{S_r}$ | Grad | 2.599687 | 2.651435 | **3.287798** | 1.778592 |
| | IG | 2.621901 | 2.686064 | **3.319311** | 1.777737 |
| Robustness-$S_r$ | Grad | 2.517404 | 2.608013 | **3.188270** | 1.750599 |
| | IG | 2.515343 | 2.603118 | **3.187848** | 1.749097 |

### 3.3.1 COMPARISON BETWEEN DIFFERENT BACKBONES

In Table 4, we show the robustness results with top 25% attacked features on the average over 400 frames randomly chosen with exactly 1 labeled triplet. On one hand, we observe that the DenseNet-121 backbone consistently outperforms other network architectures on both evaluation criteria Robustness-$S_r$ and Robustness-$\overline{S_r}$. This suggests that DenseNet-121 backbone does capture different explanation characteristics which ignored by other network backbones. On the other hand, our results are supported by the finding in (Hsieh et al., 2020), as IG performs better than Grad; and the attack on relevant features yields lower robustness than perturbing the same percentage of irrelevant features.

### 3.3.2 ROBUSTNESS EXPLANATION FOR SPECIFIC IMAGES

To more objectively evaluate the robustness explanation for specific images, we show: (a) Visualisation of important features, (b) Robustness-$\overline{S_r}$, (c) Robustness against the percentage of Top features, and (d) Robustness-$S_r$ in Figure 3. In Figure 3 (a), we visualise the Top 15% features (with yellow dots) by Grad and IG, respectively, and overlay it on manually labelled region containing instrument (in red) and target (in green). We observe that the best performed backbone (can be seen from the robustness comparison curves in Figure 3 (c)) on the specific image is the one that not only pays attention to *core attributes, but also the spurious attribute.* In the image VID08-000188, the best performed model is ResNet-18, which shows the ambiguous condition on individual images. In a closer look at Figure 3 (a), a small portion of the most relevant feature extracted by ResNet-18 is

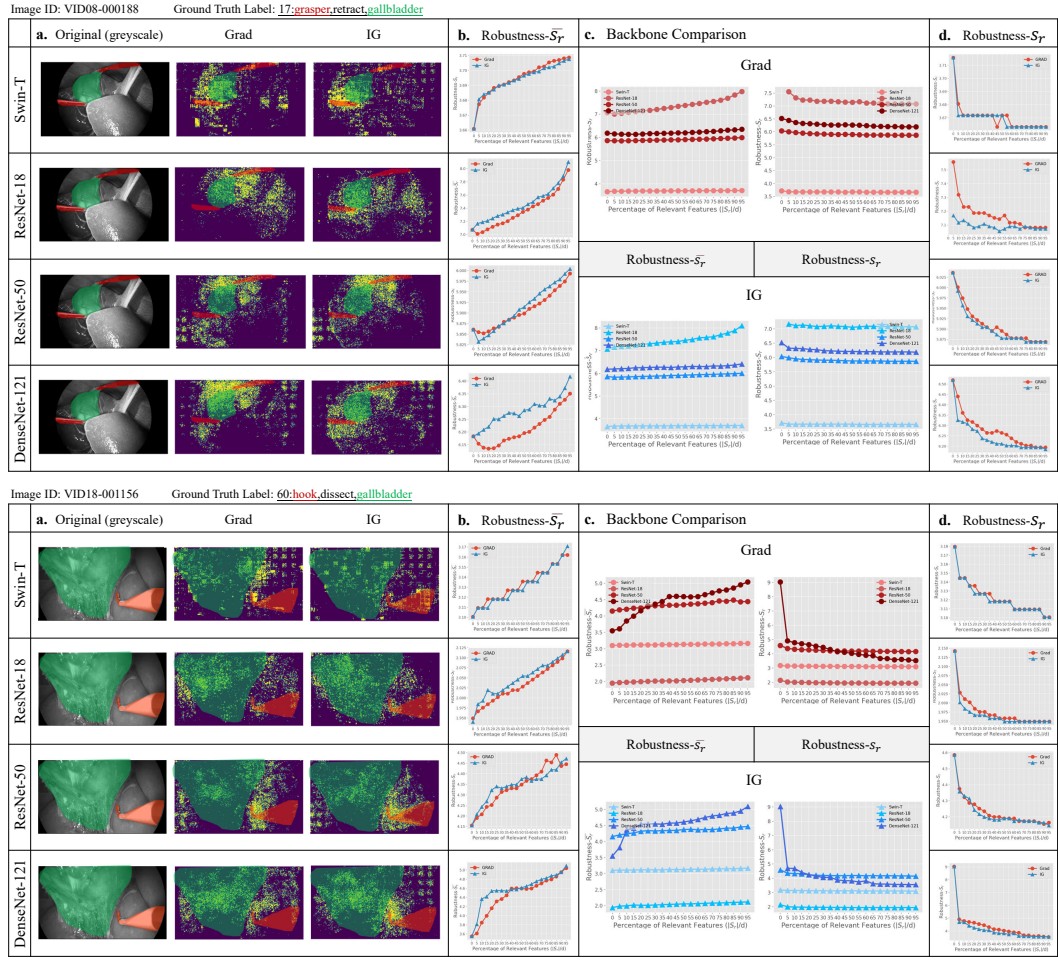

Figure 3: The set of figures shows robustness analysis on randomly selected images with a. the visualisation of the Top 15 percent of important features selected by the 2 explanation methods-Grad and IG; b. (/d.) the trends showing the robustness measured on the relevant $S_r$ (/irrelevant $\overline{S_r}$) features been selected by the 2 explanation methods against the percentage of Top features been defined as relevant; c. the comparison of the robustness across the 4 backbones embedded in Rendezvous baseline.

spread not on the close surrounding of the object area. This importance of spurious attribute is further highlighted in image VID18-001156. We observe that DenseNet-121 provides the most robust result highlighting relevant features within the tissue region and across tool tip. The worst performed model—ResNet-18 merely treated the core attributes as relevant.

The relevant role of spurious attributes can be explained by the nature of the triplet, which consists a verb component that is not the physical object. Overall, we observe that reliable deep features are the key for robust models in triplet recognition. Moreover, we observe, unlike existing works of robustness against spurious features, that both core and spurious attributes are key for the prediction.

## 4 CONCLUSION

We present the first work to understand the failure of existing deep learning models for the task of triplet recognition. We provided an extensive analysis through the lens of robustness. The significance of our work lies on understanding and addressing the key issues associated with the substantially limited in performance of existing techniques. Our work offers a step forward to more trustworthy and reliable models.

## ACKNOWLEDGEMENTS

YC and AIAR greatly acknowledge support from a C2D3 Early Career Research Seed Fund and CMIH EP/T017961/1, University of Cambridge. CBS acknowledges support from the Philip Leverhulme Prize, the Royal Society Wolfson Fellowship, the EPSRC advanced career fellowship EP/V029428/1, EPSRC grants EP/S026045/1 and EP/T003553/1, EP/N014588/1, EP/T017961/1, the Wellcome Innovator Awards 215733/Z/19/Z and 221633/Z/20/Z, the European Union Horizon 2020 research and innovation programme under the Marie Skodowska-Curie grant agreement No. 777826 NoMADS, the Cantab Capital Institute for the Mathematics of Information and the Alan Turing Institute.

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
