# OpenReview forum: "Why Deep Surgical Models Fail?: Revisiting Surgical Action Triplet Recognition through the Lens of Robustness"
_ICLR.cc/2023/Workshop/TML4H — ICLR 2023 Workshop TML4H Poster_

### Official Review · Reviewer_DHfq · 2023-03-01
**The results of this study can not generally answer the question raised in the title.**

**Rating:** 6
**Confidence:** 4

**Review:**

The paper focuses on the task of "surgical action triplet recognition" and hypothesizes that: the low performance of the existing deep learning methods arises from the ineffectiveness of the extracted features. To support this argument, the study conducted a variety of experiments and provide rich empirical results including: 1. comparing the performances of baseline models using different backbone networks as the feature extractors; and, 2. examining the robustness of the learned deep features by measuring their resistance to the perturbation under adversarial attack training. They claim their contribution as being the first study to explore the reason behind the failure of deep learning models in this task.

The good things about this paper are:
1. it well introduces the topic and made an in-depth analysis of the existing methods;
2. the study conducted generous experiments with relatively fair comparisons and considerable data and results. Some of them, such as the robustness test under adversarial attacks, indeed revels some potential areas of improvement in the existing methods of this task.

The following points may need to be improved:
1. the study only focuses on the task of "surgical action triplet recognition" which may not generally reflect all "deep surgical models";
2. there can be a variety of reasons that contributes to the poor performance of deep learning models and the unreliability of the extracted deep features may not be the only answer to the question of why deep surgical models fail;
3. the analysis and conclusion in this paper (Section 3.3) tend to be made by intuition rather than based on its experimental data or empirical evidence, and the provided experimental results may not rigorously prove the hypothesis that "the failure of the existing deep learning model in this task results from the lack of reliability in the deep features";
4. all the baseline methods used in this study originate from nearly the same groups of authors, which is not very common and may not generally reflect the common problems of this field.

---

### Meta-Review · Area_Chair_6wcA · 2023-03-03

**Recommendation:** Accept (Poster)
**Confidence:** 4

**Metareview:**

The authors study the robustness of the surgical action triplet recognition algorithms and verified the importance of feature extraction. The paper is well written. The topic fits into the workshop well. It is intuitive to use adversarial learning to study the impact of different components. The finding that both core and spurious features are important is valuable to the community.
One concern is there are many factors that may affect the performance and robustness of the algorithm. Feature extraction is important but is not the only one. The failure of deep surgical model could be due to various reasons.